# Digital morphometry and cluster analysis identifies four types of melanocyte during uveal melanoma progression

Gustav Stålhammar [1,2] & Viktor Torgny Gill [1,3]

## Abstract

**Background** Several types of benign and malignant uveal melanocytes have been described based on their histological appearance. However, their characteristics have not been quantified, and their distribution during progression from normal choroidal melanocytes to primary tumors and metastases has not been reported.

**Methods** A total of 1,245,411 digitally scanned melanocytes from normal choroid, choroidal nevi, primary uveal melanomas, and liver metastases were entered into two-step cluster analyses to delineate cell types based on measured morphometric characteristics and expression of protein markers.

**Results** Here we show that a combination of the area and circularity of cell nuclei, and BAP-1 expression in nuclei and cytoplasms yields the highest silhouette of cohesion and separation. Normal choroidal melanocytes and three types of uveal melanoma cells are outlined: Epithelioid (large, rounded nuclei; BAP-1 low; IGF-1R, IDO, and TIGIT high), spindle A (small, elongated nuclei; BAP-1 high; IGF-1R low; IDO, and TIGIT intermediate), and spindle B (large, elongated nuclei; BAP-1, IGF-1R, IDO, and TIGIT low). In normal choroidal tissue and nevi, only normal melanocytes and spindle A cells are represented. Epithelioid and spindle B cells are overrepresented in the base and apex, and spindle A cells in the center of primary tumors. Liver metastases contain no normal melanocytes or spindle A cells.

**Conclusions** Four basic cell types can be outlined in uveal melanoma progression: normal, spindle A and B, and epithelioid. Differential expression of tumor suppressors, growth factors, and immune checkpoints could contribute to their relative over- and underrepresentation in benign, primary tumor, and metastatic samples.

### Plain language summary

In this study, we take a close look on more than a million cells of the type that makes pigment inside the eye. Sometimes these cells become cancers called uveal melanomas, that have a high risk of killing the patient. The cells were digitally scanned and we measured their size, shape, and content of different proteins that are important for how they behave. We find that four types of cells are present in different proportions in normal tissue, moles, eye tumors, and tumor tissue that has spread to other parts of the body. This knowledge is important as it improves our understanding of what cells form uveal melanomas, and of how they are distributed over different stages of the disease. In turn, this could help researchers focus on the right type of cells in our pursuit of effective treatments.

[1] Department of Clinical Neuroscience, Division of Eye and Vision, Karolinska Institutet, Stockholm, Sweden. [2] St. Erik Eye Hospital, Stockholm, Sweden. [3] Department of Pathology, Vastmanland Hospital, Vasteras, Sweden. ✉email: gustav.stalhammar@ki.se

Uveal melanoma is the most common primary intraocular malignancy in adults, with an estimated global incidence of over 7000 cases per year[1]. A large proportion of patients develop metastases, and 15 years from diagnosis the relative survival in uveal melanoma is 60%[2]. Uveal melanomas originate from neural crest-derived melanocytes that normally have spindle-shaped nuclei and pigmented cytoplasms[3]. In malignant transformation, uveal melanoma tumor cells may adopt different size and shapes. In 1931, Callender described six different cell types based on histological appearance[4]. This classification has been modified in later periods and most current ophthalmic pathologists now recognize at least two basic cell types: Spindle cells (elongated cigar shaped cells with small nuclei) and epithelioid cells (large rounded cells with a higher prevalence of nucleoli)[5]. However, there is still variability in how uveal melanoma histology is interpreted, and distinctions are based on visual assessments of the shape, size, and staining characteristics of cells, which is associated with a degree of subjectivity. Typically, no classifications based on quantifiable or measured qualities are used. Many ophthalmic pathologists separate spindle A (smaller, with fine nuclear chromatin and indistinct nucleoli) from spindle B cells (larger, with coarser chromatin and distinct nucleoli). A smaller type of epithelioid cell has also been described, with less cytoplasm and a smaller nucleus than the epithelioid cell type described by Callender[5]. Further, even though epithelioid cells have been associated with an increased risk for metastasis, there is still a lack of correlation between the tumor cell types and other prognostic factors including loss of expression of BRCA1 associated protein 1 (BAP-1), which is found in a vast majority of metastasizing uveal melanomas; insulin-like growth factor 1 receptor (IGF-1R) that is synthesized in the liver and may contribute to metastatic progression; and immune checkpoint receptors indoleamine 2,3-dioxygenase (IDO) and T cell Ig and ITIM domain (TIGIT), which are overexpressed in metastasizing uveal melanoma[6–20].

In order to outline cell types based on objective measurements, to determine their protein expression characteristics, and their distribution in early and late stages of metastatic progression, we perform digital morphometry and cluster analysis of more than one million normal and malignant uveal melanocytes. We measure the expression of BAP-1, IGF-1R, IDO, and TIGIT in the identified cell types, and examine their spatial distribution in normal choroid, nevi, primary tumors, and liver metastases. As a result, we identify a total of four distinct cell types. These have a similar morphological appearance to what has been described as normal melanocytes, spindle A, spindle B, and epithelioid cells in histological studies. We also demonstrate the complete absence of spindle A cells in metastases, which indicates that they are devoid of metastatic potential.

## Methods

**Patients and samples.** Enucleated eyes and liver metastases were identified in the archive of the St. Erik Ophthalmic Pathology Laboratory. Inclusion criteria were: (1) Histologically proven melanoma in the choroid and/or ciliary body (iris melanomas were not considered), (2) Paraffin block with enucleated eye or tissue from liver metastasis available, (3) Sufficient tissue for routine staining and immunohistochemistry. Exclusion criteria were: (1) Patient still alive, (2) Tissue older than 40 years, (3) Tissue fragmented or affected by artefacts, (3) (for primary tumors) History of plaque brachytherapy, proton beam irradiation and/or transpupillary thermotherapy (TTT) prior to enucleation, or (4) (for metastases) History of systemic chemotherapy, liver perfusion therapy (isolated hepatic perfusion, percutaneous hepatic perfusion or similar), immunotherapy,

external beam irradiation or any other non-surgical treatment prior to tissue excision. The reason for exclusion criterion 3 and 4 is that such treatment may affect tumor histology. Normal choroidal melanocytes and choroidal nevi in the enucleated eyes were analyzed on the condition that they were separated from the tumor by at least 3 mm of histologically normal tissue, and were not affected by inflammation, edema, necrosis, bleeding, artefacts, or similar. The study adhered to the tenets of the Declaration of Helsinki. Methods were carried out in accordance with relevant guidelines and regulations. The protocol for collection of specimens from St. Erik Eye Hospital, Stockholm, Sweden was approved by the Swedish Ethical Review Authority (reference 2020-02172). Informed consent was waived on the condition that only deceased patients were included.

**Immunohistochemistry.** Paraffin blocks were cut into 4 μm sections, pretreated in EDTA-buffer at pH 9.0 for 20 min and incubated with mouse monoclonal antibodies against BAP-1 (clone C-4, catalog no. sc.28383, Santa Cruz Biotechnology, Dallas, Texas, USA), IDO (catalog no. 05-840, Sigma-Aldrich, Saint Louis, MO, USA), and rabbit monoclonal antibodies against IGF-1R (catalog no. ab39398, Abcam, Cambridge, UK), and TIGIT (catalog no. ab233404, Abcam). A red chromogen was used, and sections were counterstained with hematoxylin and rinsed with deionized water. The deparaffinization, pretreatment, primary staining, secondary staining and counterstaining steps were run in a Bond III automated IHC/ISH stainer (Leica, Wetzlar, Germany). Dilutions between 1:20 and 1:1500 had been evaluated by a pathologist (GS), before selecting 1:40 for BAP-1, 1:1000 for IGF-1R and 1:75 for IDO and TIGIT. For assessment of BAP-1 status in a light microscope, we followed a previously used classification[7,9]. The area in a lesion with the most intense nuclear BAP-1 staining was selected under low-power magnification (×40). The percentage of melanocytes with any visible immunoreactivity above background was then assessed in 3 high-power fields (×200). Lesions with ≥33% immunoreactive cells were classified as BAP-1 positive, and lesions with <33% immunoreactive cells as BAP-1 negative.

**Digital image analysis.** After sectioning and staining, all glass slides were digitally scanned at ×400 (Ocus 40, Grundium Oy, Tampere, Finland). Analyses of cell morphometric features and IHC expression levels were performed with QuPath Bioimage analysis v. 0.3.2[21]. The software was run on a standard off-the-shelf desktop computer (Apple Inc. Cupertino, CA).

For assessment of the levels of BAP-1, IGF-1R, IDO, and TIGIT expression, one positive cell and one negative cell, as manually selected by a pathologist (GS), was calibrated in each digitally scanned tissue section. This adjusts for differences in general staining intensity and color nuance between different slides. A polygon was then drawn around the margins of the tumor. Areas with inflammatory infiltrates, bleeding, heavy pigmentation, necrosis, poor fixation, uneven staining, and artefactual folding were excluded. Each section was also evaluated in high magnification to exclude any areas with tumor-infiltrating lymphocytes, granulocytes, macrophages, fibroblasts, hepatocytes (in the liver), and other potentially false positive cells. Segmentation parameters were setup by a pathologist in the positive cell detection tool. Based on randomly selected areas of normal choroid, nevi, primary tumors, and metastases, a background radius of 8 μm; a sigma of 2 μm; a minimum nucleus area of 10 μm$^2$; a maximum nucleus area of 200 μm$^2$; and a cell (cytoplasm) expansion of 3.5 μm, was deemed to appropriately capture the shape and size of melanocytes. A full list of the used parameters and visual examples are provided in Supplementary

**Table 1 BAP-1 classification of analyzed tissues.**

|  | Normal choroid ($n = 7$) | Choroidal nevi ($n = 2$) | Primary tumors ($n = 17$) | Metastases ($n = 6$) |
|---|---|---|---|---|
| BAP-1 positive, $n$ (%) | 7 (100) | 2 (100) | 12 (71) | 0 (0) |
| BAP-1 negative, $n$ (%) | 0 (100) | 0 (0) | 5 (29) | 6 (100) |

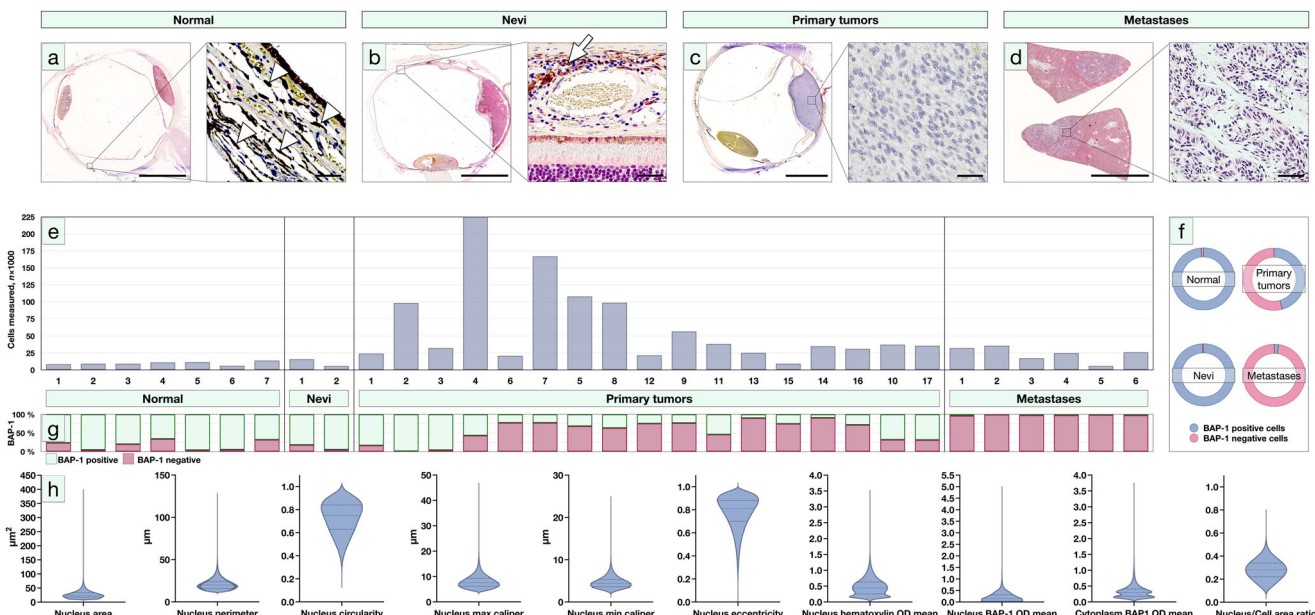

**Fig. 1 Measurement of morphometric features and protein expression in benign and malignant choroidal melanocytes.** Examples of the appearance of (**a**) normal choroidal melanocytes (arrowheads), (**b**) a choroidal nevus (arrow), (**c**) a primary uveal melanoma, and (**d**) a uveal melanoma metastasis in the liver. **e** Bar plot of the number of melanocytes analyzed in normal choroidal tissue, nevi, primary tumors, and metastases. **f** Proportion of nBAP-1 positive and negative cells in normal choroidal tissue, nevi, primary tumors, and metastases. **g** Proportion of nBAP-1 positive and negative cells in each examined specimen. **h** Violin plots of the distribution of measured morphometric variables over all included cells. Horizontal lines indicate median values and dashed lines quartiles. Scale bars: eyes and metastasis 5 mm. Histology: 50 μm.

Table 1 and Supplementary Fig. 1, respectively. The following morphometric and staining characteristics of all cells inside the polygon were then measured, as described previously[22]: area of the nucleus; nucleus perimeter; nucleus circularity (calculated as four times π times the area divided by the perimeter squared. The circularity of a circle is 1.00, and less for less circular objects); nucleus maximum caliper; nucleus minimum caliper; nucleus eccentricity (a measure of how much the nucleus deviates from a spherical shape. A completely spherical nucleus has an eccentricity of 0.00, a nucleus with the shape of an elliptical 3D solid would have an eccentricity of 0.5, whereas a 3D conical distribution has a value of 1.00); nucleus hematoxylin mean optical density (OD), nuclear BAP-1 (nBAP-1) mean OD, cytoplasm BAP-1 mean OD, and the nucleus to cell area ratio. IGF-1R expression levels were measured in cell membranes, and IDO and TIGIT in cytoplasms. A full list with a description of the analyzed morphometric and staining characteristics is provided in Supplementary Table 2. Morphometric analyses were performed blinded to all other patient data including outcome.

**Statistics and reproducibility.** We performed two-step cluster analyses to delineate cell types based on morphometric characteristics and expression of protein markers. A log-likelihood distance measure and the Schwarz's Bayesian Criterion (BIC) for model selection were applied (a criterion for model selection that introduces a penalty for the number of parameters to avoid overfitting). The maximum number of possible clusters was set to

6, considering the number of cell types in the original classification by Callender[4]. The combination of morphometric variables that resulted in the highest possible silhouette measure of cohesion and separation (i.e., how similar an object is to its own cluster compared to other clusters) were to be selected for definition of the number and characteristics of different tumor cell types. For tests of continuous variables that did not deviate significantly from a normal distribution (Shapiro–Wilk test $p > 0.05$) Student's $t$ tests were used. For non-parametrical data, Mann–Whitney $U$ tests were used. For comparisons of continuous variables across three categories or more, we used Kruskal–Wallis tests. In comparisons of two categorical variables, we used contingency tables and Pearson chi-square ($\chi^2$) tests. $p$ values below 0.05 were considered statistically significant, all $p$ values being two-sided. All statistical analyses were performed using IBM SPSS statistics version 27 (Armonk, NY, USA) and GraphPad Prism version 9.3.0 (San Diego, CA, USA).

**Reporting summary.** Further information on research design is available in the Nature Portfolio Reporting Summary linked to this article.

## Results

**Descriptive statistics.** In total, 57,255 normal choroidal melanocytes from 7 enucleated eyes, 18,276 melanocytes from choroidal nevi in 2 enucleated eyes, 1,028,086 tumor cells from 17 primary tumors, and 141,794 tumor cells from 6 metastases were

| Table 2 Morphology and staining characteristics of BAP-1 positive and negative primary tumors. | | | |
|---|---|---|---|
| | **BAP-1 negative primary tumors** | **BAP-1 positive primary tumors** | $p^a$ |
| | **Mean (SD)** | **Mean (SD)** | |
| Area of the nucleus, $\mu m^2$ | 27.81 (14.42) | 24.62 (13.81) | <0.001 |
| Nucleus perimeter | 21.63 (6.53) | 20.63 (6.53) | <0.001 |
| Nucleus circularity | 0.74 (0.14) | 0.72 (0.14) | <0.001 |
| Nucleus maximum caliper | 8.17 (2.51) | 7.74 (2.46) | <0.001 |
| Nucleus minimum caliper | 4.72 (1.40) | 4.49 (1.33) | <0.001 |
| Nucleus eccentricity | 0.77 (0.14) | 0.78 (0.14) | 0.74 |
| Nucleus hematoxylin mean OD | 0.59 (0.25) | 0.34 (0.22) | <0.001 |
| nBAP-1 OD | 0.04 (0.12) | 0.21 (0.19) | <0.001 |
| Cytoplasm BAP-1 OD | 0.25 (0.14) | 0.23 (0.33) | <0.001 |
| Nucleus to cell area ratio | 0.30 (0.08) | 0.26 (0.09) | <0.001 |

*SD* standard deviation, *nBAP-1* nuclear BAP-1 expression, *OD* optical density.
[a]Mann–Whitney *U* test.

analyzed. Both nevi were analyzed in choroids that also contained a melanoma, and the total number of cells analyzed were 1,245,411, in 32 specimens from 30 patients.

On the specimen level, all normal choroids (100%), both choroidal nevi (100%), 12 of 17 primary tumors (71%), and none of the metastases (0%) were classified as BAP-1 positive by examination in a light microscope ($\chi^2 < 0.001$, Table 1 and Fig. 1a–d). In digital image analysis, 99% of melanocytes in normal choroidal tissue and nevi were nBAP-1 positive. Of melanocytes in primary tumors and metastases, only 46 and 2% of were nBAP-1 positive, respectively (Fig. 1e–g). The distribution of the proportion of nBAP-1 positive cells differed between the four tissue types (Kruskal–Wallis $p < 0.001$). On average, the area of all tumor cell nuclei, regardless of tissue, was 26.3 $\mu m^2$ (standard deviation, SD, 14.2), and the mean nucleus circularity 0.73 (SD 0.14). The mean nucleus to cell area ratio was 0.28 (SD 0.09, Fig. 1g).

There were significant differences in all examined morphometric and staining characteristics between normal choroidal melanocytes, nevi, primary tumors, and metastases (Kruskal–Wallis $p < 0.001$, Supplementary Table 3). nBAP-1 negative cells had a higher nucleus to cell area ratio than nBAP-1 positive cells (median 0.30 vs. 0.26, Mann–Whitney *U* $p < 0.001$). nBAP-1 negative cells also had greater nuclear perimeters, circularity, maximum and minimum nucleus calipers, and nucleus hematoxylin OD. The only parameter that did not differ between BAP-1 positive and negative cells was nucleus eccentricity ($p = 0.74$, Table 2). In linear regressions, the area of the nucleus was negatively correlated to the level of BAP-1 expression in both nuclei and cytoplasms (Supplementary Fig. 2).

**Clustering**. In two-step clustering of primary tumor melanocytes, the highest silhouette measure of cohesion and separation was obtained with nucleus area, nucleus circularity, nBAP-1 OD, and cytoplasm BAP-1 OD (0.38). Nucleus perimeter, nucleus maximum caliper, nucleus minimum caliper, nucleus eccentricity, nucleus hematoxylin mean OD, or the nucleus to cell area ratio did not improve the model. Three tumor cell clusters were identified: the first consisted of nBAP-1 negative cells with large, rounded nuclei (45% of the sample). Morphologically, this cell type was similar to what has been described as the epithelioid cell type in early histological studies. The second consisted cells with higher BAP-1 expression in both nuclei and cytoplasms, with small, elongated nuclei resembling spindle A cells (23% of the sample). The third cell type had very large and elongated nuclei, which resembled spindle B cells, and had low expression levels of BAP-1 in both nuclei and cytoplasms (32% of the sample,

Fig. 2a–d). Differences in median nucleus area, nucleus circularity, nBAP-1 OD, and cytoplasm BAP-1 OD were significant between the three cell types (Kruskal–Wallis $p < 0.001$, Fig. 2f). In total, 0.2% of the analyzed cells had small, round nuclei with high BAP-1 expression, likely representing a population of tumor-infiltrating lymphocytes.

**Protein expression profiles of the three cell types**. Next, we examined the immunohistochemical expression of IGF-1R, IDO and TIGIT in the same set of primary tumors. By two-step clustering based on nucleus area and circularity, the same set of distinct cell types were outlined: epithelioid, spindle A, and spindle B.

Epithelioid cells had greater membranous expression of IGF-1R than spindle B cells, who in turn had greater expression than spindle A cells (Kruskal–Wallis $p < 0.001$).

Cytoplasmic expression of immune checkpoints IDO and TIGIT followed a different pattern: for both proteins, epithelioid cells had the greatest expression, but spindle A cells had greater expression than spindle B cells (Kruskal–Wallis $p < 0.001$, Fig. 2f).

**Spatial distribution of cell types**. We examined the spatial distribution of cell types. First, we applied morphometric analyses and clustering to melanocytes in normal choroidal tissue and in the two nevi (silhouette measure of cohesion and separation 0.35). This identified a fourth cell type with smaller nucleus area and lower nuclear circularity than spindle A cells, but with lower expression of BAP-1 in both nuclei and cytoplasms (Kruskal–Wallis $p < 0.001$, Supplementary Fig. 3). In examination of their histological appearance, they corresponded to normal choroidal melanocytes, which constituted 82% of the melanocytes in normal choroidal tissue, and 47% of the melanocytes in nevi. The remaining 18% of cells in normal choroid and 53% in nevi were spindle A cells.

In relation to the apical, central and basal portion of the tumors, as outlined by a pathologist (GS), epithelioid and spindle B cells were relatively more common in the tumor base and apex, but relatively uncommon in the central portion of tumors. Conversely, spindle A cells were relatively common in the central portion of tumors ($\chi^2$ $p < 0.001$).

Lastly, we repeated the morphometric analyses and clustering to the liver metastases. The median tumor cell area was larger in metastases than in primary tumors (103.5 vs. 87.5 $\mu m^2$, Mann–Whitney *U* $p < 0.001$). Similarly, the median tumor cell nucleus was larger in metastases than in primary tumors (23.0 vs. 22.0 $\mu m^2$, Mann–Whitney *U* $p < 0.001$). In cluster analysis based on nucleus area, nucleus circularity, nBAP-1 OD, and cytoplasm

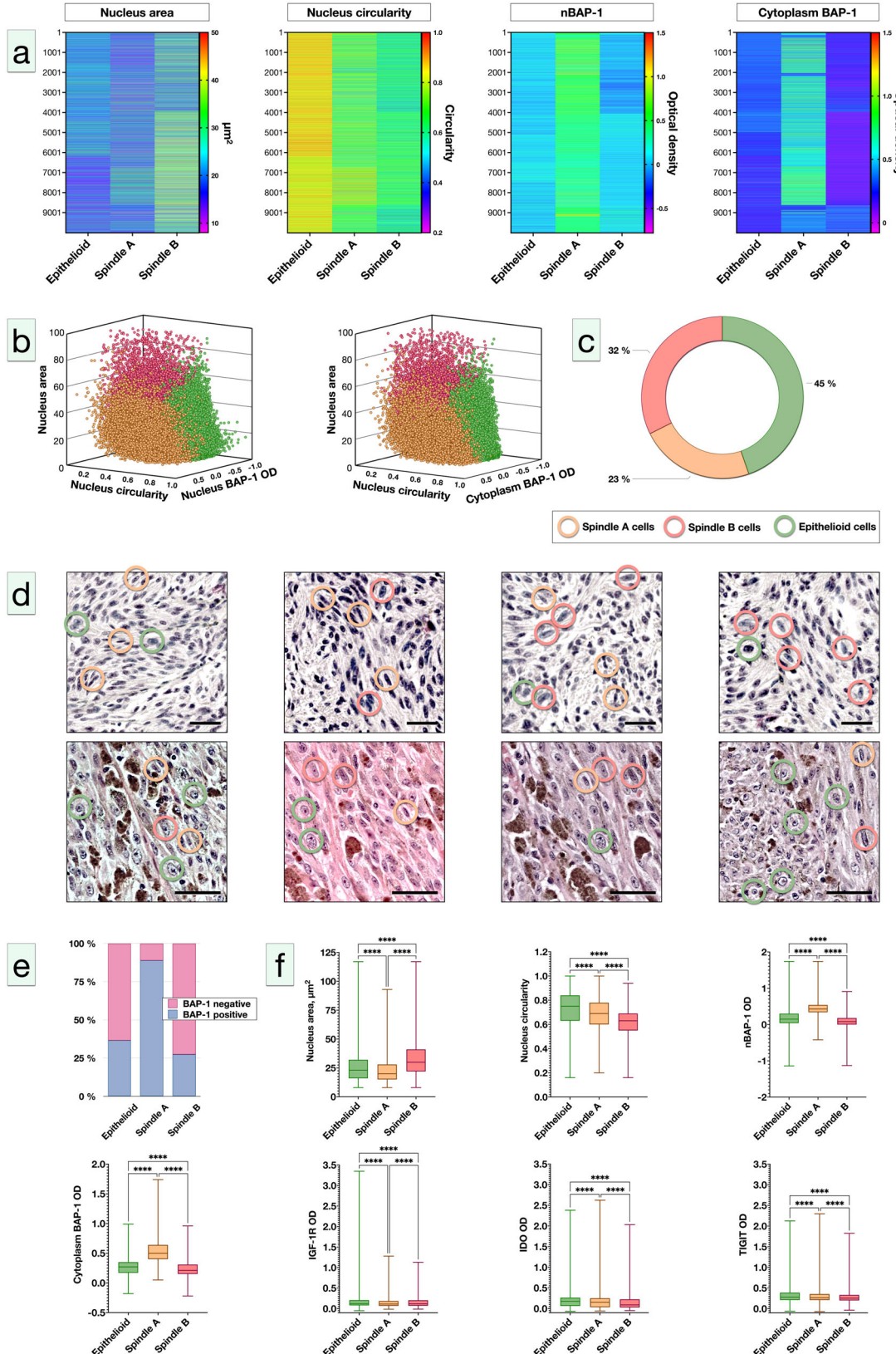

BAP-1 OD, an even higher silhouette measure of cohesion and separation was obtained (0.50). In metastases, only two of the three cell types were represented: the epithelioid and the spindle B type. Epithelioid cells constituted 69% of metastatic tumor cells, and spindle B cells 31%. In qualitative assessment of the distribution of cell types in metastases, it was noted that spindle

B cells were more common in the liver sinusoids and portal areas, whereas large aggregates of extra-sinusoidal metastatic cells more often represented the epithelioid cell type (Supplementary Fig. 4). The primary growth pattern was nodular, with elements of desmoplastic and pushing growth. Mitoses were rather rare in both cell types: In average, less than 1 mitosis was found per 5

**Fig. 2 Results of cluster analyses. a** Heat maps of nucleus area, nucleus circularity, nBAP-1 optical density (OD), and cytoplasm BAP-1 OD across the tumor cell types. This combination achieved the highest silhouette measure of cohesion and separation. Heat maps display 10,000 cells of each type. **b** 3D scatter plot of epithelioid cells (green), spindle A cells (yellow), and spindle B cells (green) across the morphometric and protein expression variables. **c** Distribution of cell types in 17 primary tumors. **d** Examples of epithelioid cells (green circles), spindle A cells (yellow circles), and spindle B cells (red circles). **e** Distribution of BAP-1 status in epithelioid, spindle A, and spindle B cells. **f** Box plots of the distribution of nucleus area, nucleus circularity, nBAP-1 OD, cytoplasm BAP-1 OD, IGF-1R OD, IDO OD, and TIGIT OD. Centerlines denote median values, while the boxes contain the 25th to 75th percentiles of the datasets. Whiskers indicate minimum and maximum values. ****Kruskal–Wallis $p < 0.0001$. nBAP-1 nuclear expression of BAP-1, OD optical density. Scale bars: 50 µm.

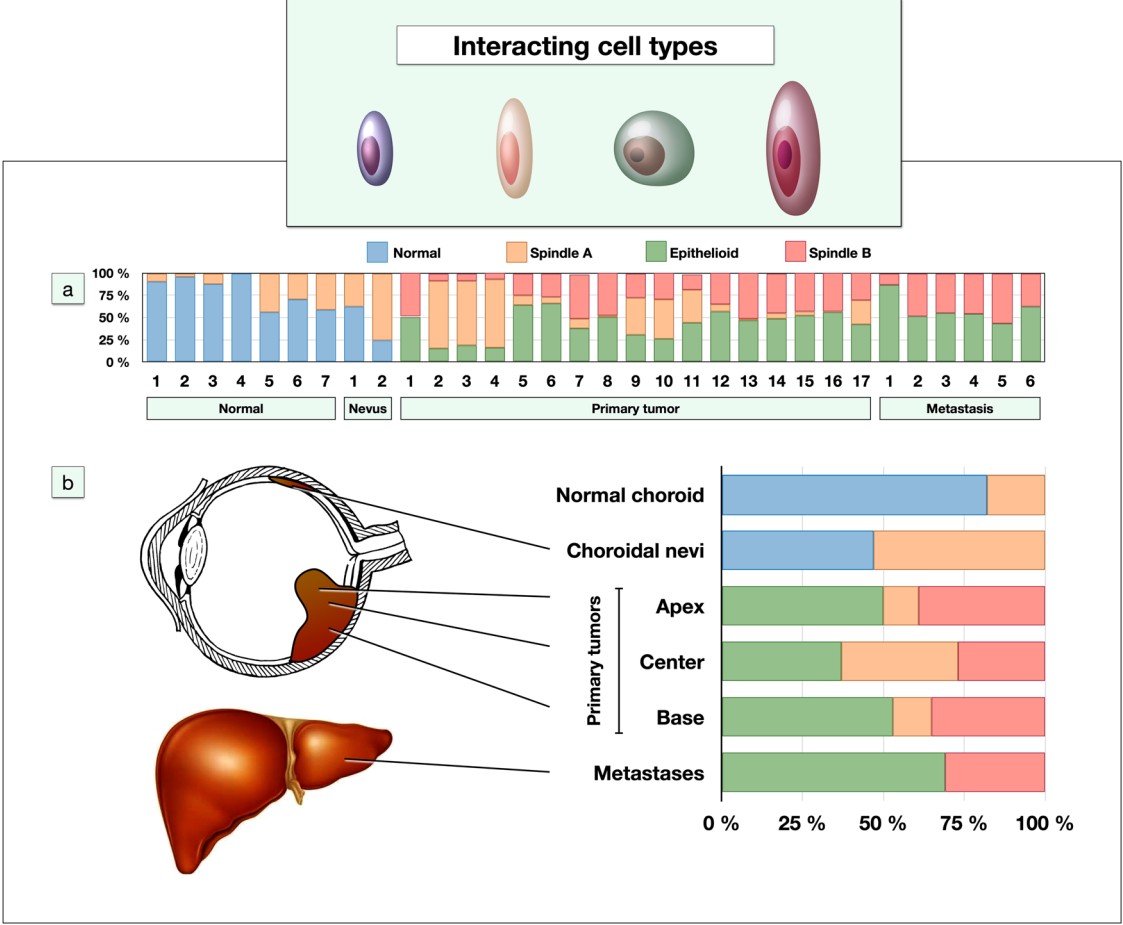

**Fig. 3 Distribution of cell types in normal choroid, choroidal nevi, primary tumors, and metastases. a** Proportions of normal, spindle A, spindle B, and epithelioid cells in each examined specimen. **b** Proportions of cell types in normal choroidal tissue; nevi; base, center and apex of primary tumors; and in metastases.

**Table 3 Characteristics of the four cell types.**

|  | Normal Mean (SD) | Epithelioid Mean (SD) | Spindle A Mean (SD) | Spindle B Mean (SD) | $p^a$ |
|---|---|---|---|---|---|
| Area of the nucleus | 21.80 (12.13) | 23.10 (10.44) | 22.38 (9.89) | 33.07 (15.52) | <0.001 |
| Nucleus circularity | 0.62 (0.16) | 0.84 (0.06) | 0.68 (0.13) | 0.61 (0.10) | <0.001 |
| nBAP-1 OD | 0.22 (0.15) | 0.11 (015) | 0.46 (0.19) | 0.08 (0.14) | <0.001 |
| Cytoplasm BAP-1 OD | 0.28 (0.17) | 0.27 (0.12) | 0.55 (0.21) | 0.23 (0.11) | <0.001 |
| Membrane IGF-1R OD | – | 0.15 (0.10) | 0.14 (0.09) | 0.14 (0.09) | <0.001 |
| Cytoplasm IDO OD | – | 0.19 (0.17) | 0.17 (0.16) | 0.14 (0.16) | <0.001 |
| Cytoplasm TIGIT OD | – | 0.31 (0.14) | 0.29 (0.12) | 0.27 (0.12) | <0.001 |
| % of primary tumor cells | 0 | 45 | 23 | 32 | <0.001 |
| % of metastatic cells | 0 | 69 | 0 | 31 | |

*SD* standard deviation, *nBAP-1* nuclear BAP-1 expression, *OD* optical density.
[a]Kruskal–Wallis test for continuous variables, chi-square for tumor cell distributions.

high-power fields of metastatic tissue. No normal choroidal melanocytes were found in primary tumors or metastases. Spindle A cells were found in primary tumors, but not in metastases. Consequently, the distributions of the proportion normal melanocytes (Kruskal–Wallis $p < 0.001$), Spindle A cells ($p = 0.002$), Spindle B cells ($p < 0.001$), and epithelioid cells ($p < 0.001$) differed between choroidal tissue, nevi, primary tumors, and metastases (Fig. 3a).

In summary, the normal choroidal population of melanocytes, nevi, melanomas and metastases consist of four principal cell types: normal, spindle A, spindle B, and epithelioid cells. In normal choroidal tissue and nevi, only normal melanocytes and spindle A cells are represented. The latter is also represented in primary tumors, along with epithelioid cells and spindle B cells. Epithelioid cells have large, rounded nuclei with low nuclear and cytoplasmic expression of BAP-1, and relatively high expression of IGF-1R and immune checkpoints IDO and TIGIT (i.e., high in relation to the expression in the other cell types). Spindle A cells have small, spindle-shaped nuclei with high nuclear and cytoplasmic expression of BAP-1, relatively low level of IGF-1R expression, and intermediate expression levels of IDO and TIGIT to epithelioid and spindle B cells. Spindle B cells have very large, spindle-shaped nuclei with low nuclear and cytoplasmic expression of BAP-1, and lower level of IGF-1R, IDO and TIGIT expression. Liver metastases contain no normal melanocytes or spindle A cells, which are likely devoid of metastatic potential (Table 3 and Fig. 3c).

## Discussion

In this study, we demonstrate that primary uveal melanomas consist of three tumor cell types of which only two are present in liver metastases. The findings were based on digital measurements of more than a million tumor cells. In contrast to most previous literature on uveal melanoma tumor cell morphology and staining patterns, the data produced in this project were not influenced by subjective assessments from one or several pathologists, but by objective and automated analysis. This approach has previously been shown to outperform manual assessments in terms of reproducibility and prognostic utility[23–27].

Spindle A cells have previously been associated with a more favorable prognosis[12, 15,28]. In the present study, we outline the shape and area of their nucleus, and demonstrate that they have a relatively high expression of BAP-1 in both the nucleus and cytoplasm. We demonstrate that spindle A cells are more common in the central areas of primary uveal melanomas, and less common toward the tumors' apex and base, and that they are completely absent in metastases. We also find that they have lower expression of IGF-1R and intermediate expression of immune checkpoint receptors IDO and TIGIT. In contrast, it is possible that the higher expression of IDO and TIGIT in epithelioid cells improves their ability to withstand immune-mediated eradication[29,30]. This could be an explanation to their relative abundance in metastases. The spindle B cell type has received less attention. We now demonstrate that they, together with epithelioid cells, represent the vast majority of cells in the base of primary tumors, which is also the area from which the tumor receives its vascular supply and the route for hematogenous dissemination. This is further corroborated by a previous publication, in which we found that the proportion of nBAP-1 positive tumor cells is lower along the scleral margin (i.e., base) of uveal melanomas[31].

Diffuse cytoplasmic BAP-1 expression has previously been identified in a majority of nBAP-1 positive uveal melanomas[32]. This is in line with the higher cytoplasmic BAP-1 OD that we

identified in nBAP-1 positive spindle A cells. Whereas nBAP-1 is a very strong prognostic factor that correlates with several other predictors of metastasis, the level of cytoplasmic BAP-1 expression has a weaker association with prognosis as well as prognostic subclasses based on gene-expression profiling[7,9,18,19]. Further, in 2014, Kalirai et al. found a complete concordance of nBAP-1 classification in 5 paired primary tumors and metastases, and that 10 of 13 liver metastases were nBAP-1 negative[18]. In another study from 2016, Grossniklaus et al. reported that 8 of 15 examined metastases were nBAP-1 negative, whereas Agin et al. found that all 8 examined liver metastases were nBAP-1 negative in 2022[33,34]. This highlights that although BAP-1 is a very strong prognostic indicator, it's presence or absence varies between studies, and that other factors can drive the metastatic progression of uveal melanoma. For example, SF3B1 mutations can be found in 10 to 21% of primary uveal melanomas, and identify a group of patients with late onset metastases[35–37]. Several other characteristics are involved in uveal melanoma progression and regulation of the tumor microenvironment. In this study, we had no access to genetic analyses such a single-cell exome sequencing or similar of the different cell types. This would be highly interesting to explore in future projects.

This study has at least two other limitations: the very large number of cells analyzed increases the chance that even small differences between the cell types become statistically significant. The demonstrated differences in IGF-1R expression may not be reproducible in smaller cohorts. Secondly, even though the number of analyzed cells was very high, they were distributed over a limited number of tissue specimens ($n = 32$) from an even smaller number of patients ($n = 30$). Thus, the number of technical replicates was very high but the number of biological replicates very low. This warrants caution in interpretation of our results.

In conclusion, based on morphometric characteristics and protein expression, four basic cell types can be outlined in uveal melanoma progression: normal melanocytes, spindle A, spindle B and epithelioid tumor cells. Epithelioid and spindle B cells are not found in normal choroidal tissue or in choroidal nevi, whereas spindle A cells are likely devoid of metastatic potential. Differential expression of the tumor suppressor BAP-1, growth factor receptor IGF-1R, and immune checkpoints IDO and TIGIT could contribute to the relative over- and underrepresentation of cell types at different stages from normal choroidal tissue to metastasis. Future studies should aim to correlate the morphometric- and protein expression characteristics with genetic aberrations on the single-cell level.

## Data availability

The anonymized experimental data that support the findings of this study including raw data for each compartment (normal choroidal tissue, nevi, primary tumors, and metastases), and source data for Figs. 1h, 2f and 3 are available on Figshare: https://figshare.com/projects/Digital_morphometry_and_cluster_analysis_identifies_four_types_of_melanocyte_during_uveal_melanoma_progression/164923. All other data are available from the corresponding author on reasonable request.

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

## Acknowledgements
G.S. is supported by The Swedish Cancer Society (20 0798 Fk), The Swedish Eye Foundation (2022-05-02), Karolinska Institutet (FS-2021-01131), Region Stockholm (20200356), and The Swedish Society of Medicine (SLS-971390). The funders of the study had no role in study design, data collection, data analysis, data interpretation, or writing of the report.

## Author contributions
G.S.: conceptualization, data curation, formal analysis, funding acquisition, investigation, methodology, project administration, visualization, writing—original draft, writing—review and editing. V.T.G.: validation, writing—review and editing.

## Funding

## Competing interests
The authors declare no competing interests. G.S. is an Editorial Board Member for *Communications Medicine*, but was not involved in the editorial review or peer review, nor in the decision to publish this article.
