## [Peer Review File · Communications Medicine]

Reviewers' comments:

Reviewer #1 (Remarks to the Author):

The current manuscript evaluates the characteristics of uveal melanoma cells in histological sections and attempts to quantify the distribution of cell types during disease progression from normal choroidal melanocytes, primary tumors, and metastatic dissemination. Implemented quantification methods include high-precision automated morphological analysis. The authors conclude the existence of morphological parameters associated with differential expression of nuclear BAP1, IDO, TIGIT, and IGF-1R during UM evolution.

Major comments:

The main weakness of this work is the sample size used and the statistical analysis applied to support the authors' conclusions.

Statistical analyses were performed considering the number of cells, not patients (line 158).

In Results: Line 177, the authors disclose the sample size for each group:

Normal choroid: 7 enucleated eyes

Choroidal nevi: 2 enucleated eyes

Primary tumors: 17
46% melanocytes nBAP1+

Metastatic tumors: 6
2% melanocytes nBAP1+

Comment 1: Instead of disclosing percentages of nBAP1+ melanocytes, also inform the number of patients with this phenotype for each primary and metastatic sample.

Comment 2: In Results (lines 173-186), Statistical analysis should be performed in light of the sample size and NOT solely in the light of the number of cells (technical size). For this purpose, averages or medians for each subject should be considered in a group analysis for further statistical tests. Whether the differences are significant or not across different types of tumors and samples (patient sample size, not cells), it is OK to infer the qualitative differences based on technical quantifications (especially considering that this is a rare tumor type), but the authors should always acknowledge this in the text, and that the sample size wouldn't necessarily support the conclusions and that further studies should be carried out with a larger group. For example, 2% of melanocytes in metastatic samples are nBAP1+. This information is very vague.

Is there a tumor type diagnosed as nBAP1+ at all? To compare different phenotypic

characteristics across tumors, this should be done at the level of tumor type/patient, since other characteristics governing UM disease were not considered as well, such as genetic features, and they differ and might impact the tumor microenvironment, which includes IDO, TIGIT, and other checkpoint markers expression. Therefore, this should be acknowledged in the text before publication.

Comment 3. The quantification of the IDO and TIGIT expression is vague: high, intermediate and low. How is this quantified?

Comment 4: Morphology of metastatic UM cells to the liver can vary according to their location. For example, if within the sinusoids, they are often spindle-shaped; whilst within the large metastatic nodules, they can be 'plumper' and have a more epithelioid cell morphology. How was the morphology within the 6 mUM assessed?

Comment 5: It is assumed that iris melanomas were excluded in this study: is this the case? As has been previously described, it is thought that the anterior chamber influences the morphology of the iris melanomas. If these tumours were excluded, please make this clear, and also change the title of the paper.

Comment 6: Finally, I am wondering what is new about this paper. Essentially, it is confirming the previous morphological modified Callendar classification of UM. I.e. that there are normal melanocytes, spindle A and spindle B UM cells, as well as epithelioid cells. This paper is confirming what is already known – that most spindle A UM cells are nBAP1+ and that most epithelioid UM cells are nuclear BAP1 negative. The TIGIT and IDO findings aren't really novel – see below – and are actually vague in their quantification (see comment above).

Minor comments: Literature review before acceptance for publication:

A) References on the prognostication power of nBAP-1 at the protein level were not included, just BAP1 expression association with metastasis or genetic changes. Please take a look at original articles disclosing for the first time that lack of BAP1 protein expression is associated with high risk and its utility in prognostication and use these references to improve the introduction and discussion of the manuscript.

- 1) Shah AA et al., Pathology, 2013.
- 2) Kalirai H, et al. Br J Cancer . 2014 Sep 23;111(7):1373-80.
- 3) Koopmans AE et al. Modern Pathology, 2014
- 4) Farquhar N et al. J Pathol Clin Res . 2017 Nov 13;4(1):26-38.

B) Add subtitles to all Figure components, such as Fig.1A, Fig. 2A, 2B, etc. This helps the reader to understand the results without the need to go back and forth in the text.

C) References on IDO and TIGIT expression in metastasizing uveal melanomas were not properly cited (lines 85-86). Please take a look at the first article describing IDO and TIGIT expression in both primary and liver metastasis, both at the RNA and Protein levels in nBAP1-negative tumors.

1) Figueiredo et al. J Pathol . 2020 Apr;250(4):420-439.

D) Figure 2: Units for Nucleus circularity and Optical Density should be included in the heat maps.

Reviewer #2 (Remarks to the Author):

Summary: Stålhammar and Gill used QuPath to analyze morphometric clustering of melanocytes, choroidal melanocytes, and tumor cells from 9 enucleated eyes, 17 primary tumors, and 6 metastases. Morphological characteristics and protein expression statistics calculated included area, perimeter, circularity, maximum caliper, minimum caliper, eccentricity, hematoxylin mean optical density (OD), BAP-1 mean OD, cytoplasm BAP-1 mean OD, and nucleus to cell area ratio. As a result of two-step cluster analyses, the authors found three sub-populations of tumor cells: epithelioid, spindle A, and spindle B (a fourth cell type was also found after analysis of spatial distribution of these cells). In addition to this cluster-based analysis, the authors also analyzed relative cell populations in both normal choroidal tissue and nevi. The authors demonstrate an objective method for identifying relative proportions of different cell types that are associated with negative and favorable prognosis.

Major Comments: There are a number of details that appear to be missing from the authors' description of methods which are important towards understanding the impact of these results. Firstly, the authors do not provide any example segmentation, description of segmentation procedure, or quantification of nuclear segmentation performance. Examining the violin plots in Figure 1, there appears to be a much higher amount of variation present in their dataset than would be expected for nuclei. Some example segmentations, as well as an in-depth description of quality control procedures used to eliminate false-positive nuclei are required prior to approval. Furthermore, it is not fully explained by the authors why these seemingly very different types of tissues were all included for identification of different cell types. Are the types of cells found in enucleated eyes, primary tumors, and liver metastases significantly overlapping? Did the authors perform any relative clustering analyses when only one type of tissue was used?

Minor Comments:

- The authors should provide a layman's description or table with key definitions including the types of morphometrics calculated from the nuclei, key metrics used to quantify relative clustering (silhouette, BIC, etc.), as well as some domain-specific terminology (nevus, BAP-1, etc.).
- Some real-life examples alongside the graphical depictions of the different identified cell types would help in describing the qualitative difference between these four populations. Can an observer recognize the different types by eye? Or are the differences so subtle that only a machine can identify them?

Overall: The authors present a very interesting work with a very large number of cells to draw

conclusions from. Inclusion of the aforementioned details will strengthen the quality and reproducibility of the work.

Stockholm, January 18, 2023

Dear Editor and Reviewers,

We appreciate the review and helpful comments. The manuscript has now been amended to address all raised issues. Please find our point-by-point response and descriptions of the revisions below.

Sincerely, on behalf both authors

Gustav Stålhammar
St. Erik Eye Hospital
Karolinska Institutet
Stockholm, Sweden

Comment	Author's response	Change in the Manuscript
Reviewer #1		
Initial remarks: The current manuscript evaluates the characteristics of uveal melanoma cells in histological sections and attempts to quantify the distribution of cell types during disease progression from normal choroidal melanocytes, primary tumors, and metastatic dissemination. Implemented quantification methods include high-precision automated morphological analysis. The authors conclude the existence of morphological parameters associated with differential expression of nuclear BAP1, IDO, TIGIT, and IGF-1R during UM evolution.		
The main weakness of this work is the sample size used and the statistical analysis applied to support the authors' conclusions. Statistical analyses were performed considering the number of cells, not patients (line 158).	Thank you, we agree that this is important and we have therefore added analyses on the lesion level. As reported below, we have added the number of patients with BAP-1 positive/negative phenotype for each sample, a new table displaying the BAP-1 classification of all analyzed lesions, and a bar chart to figure 3 that shows the distribution of BAP1 positive/negative cells lesion per lesion.	Rows 189 to 213 Table 1 Figure 1E and 3A
In Results: Line 177, the authors disclose the sample size for each group: Normal choroid: 7 enucleated eyes Choroidal nevi: 2 enucleated eyes Primary tumors: 17 46% melanocytes nBAP1+ Metastatic tumors: 6 2% melanocytes nBAP1+ Comment 1: Instead of disclosing percentages of nBAP1+ melanocytes, also inform the number of patients with this phenotype for each primary and metastatic sample.	Thank you. We have now: Added the number of patients with BAP-1 positive/negative phenotype for each sample. Added statistical tests of the distribution of cell types and BAP-1 negative cells across tissues (i.e. patients). Added a new table displaying the BAP-1 classification of all analyzed lesions. Added one bar chart to figure 1 and another to figure 3. These show the distribution of BAP1 positive/negative cells lesion per lesion (patient), and the distribution of cell types, respectively.	Rows 189 to 213 Table 1 Figure 1E and 3A
Comment 2: In Results (lines 173-186), Statistical analysis should be performed in light of the sample size and NOT solely in the light of the number of cells (technical size). For this purpose, averages or medians for each subject should be considered in a group analysis for further statistical tests. Whether the differences are significant or not across different types of tumors and samples (patient sample size, not cells), it is OK to infer the qualitative differences based on technical quantifications (especially considering that this is a rare tumor type), but the authors should always acknowledge this in the text, and that the sample size wouldn't necessarily support the conclusions and that further studies should be carried out with a larger group. For example, 2% of melanocytes in metastatic samples are nBAP1+. This information is very vague.	Thank you, we agree fully. In addition to the additions on rows 181-190, 194-196, 261-264, the new table 1, and the additions to figures 1E and 3A that we mentioned above, we have also added a supplementary table 1 that reports means and SD of morphometric and staining variables for each type of tissue (i.e. normal choroid, nevi, primary tumors, and metastases). Further, limited sample size is clarified in the descriptive statistics and also underlined more clearly as a limitation to the study in the discussion. In figure 1, the proportion of BAP-1 negative cells for each specimen including metastases is now displayed.	Supplementary table 1 Rows 193 to 195 Rows 338 to 341
Is there a tumor type diagnosed as nBAP1+ at all? To compare different phenotypic characteristics across tumors, this should be done at the level of tumor type/patient, since other characteristics governing UM disease were not considered as well, such as genetic features, and they differ and might impact the tumor microenvironment, which includes IDO, TIGIT, and other checkpoint markers expression. Therefore, this should be acknowledged in the text before publication.	All normal choroidal melanocytes, all choroidal nevi, 12 of 17 primary tumors (71 %), and none of the metastases were classified as BAP-1 positive by examination in a light microscope ($\chi^2=0.001$, table 1). We now acknowledge that many other characteristics are involved in uveal melanoma progression and regulation of the tumor microenvironment than the ones included here.	Rows 196 to 198 Table 1 Rows 325 to 333

Comment 3. The quantification of the IDO and TIGIT expression is vague: high, intermediate and low. How is this quantified?	Thank you for pointing this out. For IDO, TIGIT and IGF-1R, we make no classification of what is "high", "intermediate" or "low". These descriptions are all relative: "Epithelioid cells have large, rounded nuclei with low nuclear and cytoplasmic expression of BAP-1, and relatively high expression of IGF-1R and immune checkpoints IDO and TIGIT". I.e. high in relation to the expression in the other cell types. "Spindle A...intermediate levels of IDO and TIGIT" only means that it was in between the expression levels in Epithelioid and Spindle B cells. Perhaps it is a language issue on our part. We are now trying to state this more clearly by underlining "relative" and by providing an example in this section of the manuscript.	Rows 284, and 286 to 287
Comment 4: Morphology of metastatic UM cells to the liver can vary according to their location. For example, if within the sinusoids, they are often spindle-shaped; whilst within the large metastatic nodules, they can be 'plumper' and have a more epithelioid cell morphology. How was the morphology within the 6 mUM assessed?	Thank you. The measured characteristics of tumor cells included all areas of the metastases. We have now also added a qualitative assessment of the distribution of spindle B and epithelioid cells in metastases, as well as a new figure with an example of spindle B cells in a portal area.	Rows 266 to 276 Supplementary figure 3
Comment 5: It is assumed that iris melanomas were excluded in this study: is this the case? As has been previously described, it is thought that the anterior chamber influences the morphology of the iris melanomas. If these tumours were excluded, please make this clear, and also change the title of the paper.	Yes, it is correct that only melanomas in the choroid and/or ciliary body were included, and that iris melanomas were not considered. This is now clarified in the inclusion criteria.	Rows 105 to 106
Comment 6: Finally, I am wondering what is new about this paper. Essentially, it is confirming the previous morphological modified Callendar classification of UM. I.e. that there are normal melanocytes, spindle A and spindle B UM cells, as well as epithelioid cells. This paper is confirming what is already known – that most spindle A UM cells are nBAP1+ and that most epithelioid UM cells are nuclear BAP1 negative. The TIGIT and IDO findings aren't really novel – see below – and are actually vague in their quantification (see comment above).	An excellent point. We are cautious about claiming novelty and primacy. And forgive us if we are stating the obvious in the following: The original Callender classification has undergone several modifications. Originally, 6 different types of uveal melanoma were described of which only 1 was described as being composed of a mixture of spindle and epithelioid cells. Eventually, some small spindle A melanomas were reclassified as nevi, and less emphasis was put on the distinction of spindle A and B cells. Neither the modified Callender classification from 1983 nor the AJCC Cancer staging manual 8th Ed. mention what proportion of what cells should constitute a spindle cell, epithelioid or mixed melanoma. In practice, we and many other pathologists call a tumor with >90 % spindle cells a spindle cell tumor, a tumor with >90 epithelioid cells an epithelioid tumor, and anything in between a mixed cell tumor. Although a digital method was used in the modified classification from 1983 as well as in some other papers both before and after, they have generally analyzed specific features (e.g. size of nucleolus) in a much smaller number of cells. We could therefore mention, as potential novelties, that to the best of our knowledge:  1: This manuscript is the first to make a fully objective classification. We did not label the cells as spindle-like or epithelioid and then measure differences in features, but quite the opposite, measured many different features and had these clustered into four different cell types. It is both a strength and a weakness that this resulted in the same cell types that had already been identified by manual visual inspection and "human" clustering. Yes, it would have been exciting if we had identified a fully novel type of cell that hadn't been described before. But on the other hand, we also feel reassured that we ended up with a result that is supported by many previous studies, and we therefore think our results can be trusted. 2. No previous study has objectively measured the proportion of different cell types in all of a) normal choroidal tissue, b) nevi, 3) primary tumors, and 4) metastases. 3. We have found no previous publications with morphological measurements of normal choroidal melanocytes on the histological level. 4. No previous study has based its findings on such a large number of cells. A pathologist is limited to evaluating tumor cells on a scale of hundreds, not >1 million as done here. 5. No previous study has correlated the expression of all of BAP-1, IGF-1R, IDO and TIGIT to the different cell types. We think this adds some insight into what makes epithelioid and spindle B cells overrepresented as the melanoma progresses. 6. We reintroduce spindle B cells as something worth examining closer. Based on the results herein, we are not convinced that spindle B cells are very similar to spindle A cells in prognostic 	

	terms. They express low levels of BAP-1, but they may also be more vulnerable to attacks from the immune system based on their lower levels of IDO and TIGIT.	
References on the prognostication power of nBAP-1 at the protein level were not included, just BAP1 expression association with metastasis or genetic changes. Please take a look at original articles disclosing for the first time that lack of BAP1 protein expression is associated with high risk and its utility in prognostication and use these references to improve the introduction and discussion of the manuscript. 1) Shah AA et al., Pathology, 2013. 2) Kalirai H, et al. Br J Cancer . 2014 Sep 23;111(7):1373-80. 3) Koopmans AE et al. Modern Pathology, 2014 4) Farquhar N et al. J Pathol Clin Res . 2017 Nov 13;4(1):26-38.	Thank you. All these references have been added (except Farquhar, which was already included). We have expanded the discussion on BAP1 and also SF3B1 mutations.	Rows 319 to 333
Add subtitles to all Figure components, such as Fig.1A, Fig. 2A, 2B, etc. This helps the reader to understand the results without the need to go back and forth in the text.	Thank you. Figures and legends have been reworked, and the descriptive statistics section rearranged. Hopefully it is now easier to understand the results and figures. However, we have not added individual letters to all 10 violin plots in figure 1, since they are not reported individually in the legend.	Figure 1 , figure 3
References on IDO and TIGIT expression in metastasizing uveal melanomas were not properly cited (lines 85-86). Please take a look at the first article describing IDO and TIGIT expression in both primary and liver metastasis, both at the RNA and Protein levels in nBAP1-negative tumors. 1) Figueiredo et al. J Pathol . 2020 Apr;250(4):420-439.	Thank you. We have read and referenced this paper.	Reference #20
Figure 2: Units for Nucleus circularity and Optical Density should be included in the heat maps.	Units have been added. Note: Circularity is its own unit, ranging from 0 to 1.	Figure 2A

Reviewer #2
Initial remarks: Summary: Stålhammar and Gill used QuPath to analyze morphometric clustering of melanocytes, choroidal melanocytes, and tumor cells from 9 enucleated eyes, 17 primary tumors, and 6 metastases. Morphological characteristics and protein expression statistics calculated included area, perimeter, circularity, maximum caliper, minimum caliper, eccentricity, hematoxylin mean optical density (OD), BAP-1 mean OD, cytoplasm BAP-1 mean OD, and nucleus to cell area ratio. As a result of two-step cluster analyses, the authors found three sub-populations of tumor cells: epithelioid, spindle A, and spindle B (a fourth cell type was also found after analysis of spatial distribution of these cells). In addition to this cluster-based analysis, the authors also analyzed relative cell populations in both normal choroidal tissue and nevi. The authors demonstrate an objective method for identifying relative proportions of different cell types that are associated with negative and favorable prognosis.

Major Comments: There are a number of details that appear to be missing from the authors' description of methods which are important towards understanding the impact of these results. Firstly, the authors do not provide any example segmentation, description of segmentation procedure, or quantification of nuclear segmentation performance. Examining the violin plots in Figure 1, there appears to be a much higher amount of variation present in their dataset than would be expected for nuclei. Some example segmentations, as well as an in-depth description of quality control procedures used to eliminate false-positive nuclei are required prior to approval. Furthermore, it is not fully explained by the authors why these seemingly very different types of tissues were all included for identification of different cell types. Are the types of cells found in enucleated eyes, primary tumors, and liver metastases significantly overlapping? Did the authors perform any relative clustering analyses when only one type of tissue was used?	Thank you for these great comments. We agree fully. Naturally, segmentation, exclusion of false positives etc. was done, but it wasn't described very closely. A closer description of the manual assessment of melanocytes and the image segmentation process has now been added. We have also added a table with the used cell detection settings (supplementary table 1). Additionally, we provide visual examples of the segmentation process with images of primary tumor melanocytes as well as normal choroidal melanocytes (supplementary figure 1). Lastly, a description of the different morphometric variables is provided in supplementary table 2. All melanocytes analyzed in this study is a single type of cell: The choroidal melanocyte (in other words: Yes, there is significant overlap, or rather: It is all the same cell!). The different lesions included should be viewed as a continuum. Normal choroidal melanocytes can undergo changes and grow into a nevus, that in turn can undergo further changes that makes it a primary uveal melanoma. With further changes, the tumor can metastasize, primarily to the liver. Choroidal melanocytes are originally small and spindle-shaped, but undergo significant changes in shape and size during this progression. Therefore, we are not very surprised about the variation that is shown in the violin plots in figure 1.	Rows 129 to 135 Rows 147 to 156 Supplementary table 1 Supplementary figure 1 Supplementary table 2
The authors should provide a layman's description or table with key definitions including the types of morphometrics calculated from the nuclei, key metrics used to quantify relative clustering (silhouette, BIC, etc.), as well as some domain-specific terminology (nevus, BAP-1, etc.).	Thank you, we agree. A description and interpretation of all analyzed variables is now provided in supplementary table 2. Additionally, we have clarified 1) the silhouette measure of cohesion and separation (i.e., how similar an object is to its own cluster compared to other clusters), and 2) BIC. The significance of BAP-1 is now discussed closer. The role of choroidal nevi for the progression from normal choroidal melanocytes to primary tumors and metastases is quite complicated and we would prefer to not go into that discussion in this article. In short, some authors think that uveal melanomas arise from nevi that once consisted of normal melanocytes that then undergo genetic changes, whereas some think that small uveal melanomas and nevi are two different things but initially indistinguishable on both cellular and gross levels. The truth is that no one knows, perhaps both are true, but we think it is outside the scope of this article.	Supplementary table 2. Rows 173 to 175 Row 178 Row 320 to 330
Some real-life examples alongside the graphical depictions of the different identified cell types would help in describing the qualitative difference between these four populations. Can an observer recognize the different types by eye? Or are the differences so subtle that only a machine can identify them?	In addition to the examples from scanned tissues in figure 1A to 1D, and in figure 2D, and the graphical representation in figure 3B, we now also provide further examples of the visual appearance of the different cell types in supplementary figure 1A to 1F. Yes, a human observer can recognize the different cell types by eye. In fact, they have been described in various classifications for almost 100 years! Please see introduction, rows 68 to 80.	Figure 1A to 1D Figure 2D Figure 3B Supplementary figure 1 Rows 68 to 80
Overall: The authors present a very interesting work with a very large number of cells to draw conclusions from. Inclusion of the aforementioned details will strengthen the quality and reproducibility of the work.		

REVIEWERS' COMMENTS:

Reviewer #1 (Remarks to the Author):

The authors have addressed most of the points of both reviewers.

The revised text does now contain some small scattered typographical errors, which need correction, although my be picked up by the publishing team.

In the Figure 1A, the small box (which is then magnified to show the normal choroid) is not located over the choroid, rather over the detached retina. Please correct.

Stockholm, March 22, 2023

Dear reviewer #1,

We are thankful for these two comments. The manuscript has now been proofread and we have moved the small box in figure 1a.

Sincerely, on behalf both authors

Gustav Stålhammar
St. Erik Eye Hospital
Karolinska Institutet
Stockholm, Sweden

Comment	Author's response	Change in the Manuscript
Reviewer #1		
Initial remark: The authors have addressed most of the points of both reviewers		
The revised text does now contain some small scattered typographical errors, which need correction, although my be picked up by the publishing team.	The manuscript has been carefully proofread and a few typographical errors have been corrected.	
In the Figure 1A, the small box (which is then magnified to show the normal choroid) is not located over the choroid, rather over the detached retina. Please correct.	Thank you for noticing. The small box has now been moved to its correct position over the choroid.	Figure 1a.